# A Comparative Study of Composition and Soluble Polysaccharide Content between Brewer’s Spent Yeast and Cultured Yeast Cells

**DOI:** 10.3390/foods13101567

**Published:** 2024-05-17

**Authors:** Hyun Ji Lee, Bo-Ram Park, Legesse Shiferaw Chewaka

**Affiliations:** Department of Agro-Food Resource, National Institute of Agricultural Science, Rural Development Administration (RDA), Jeonju 54875, Republic of Korea; hyunji9595@korea.kr (H.J.L.); bboram27@korea.kr (B.-R.P.)

**Keywords:** brewer’s spent yeast, cell residue, soluble polysaccharides, mannan, β-glucan, superoxide dismutase

## Abstract

Yeast, crucial in beer production, holds great potential owing to its ability to transform into a valuable by-product resource, known as brewer’s spent yeast (BSY), with potentially beneficial physiological effects. This study aimed to compare the composition and soluble polysaccharide content of Brewer’s spent yeast with those of cultured yeast strains, namely *Saccharomyces cerevisiae* (SC) and *S*. *boulardii* (SB), to facilitate the utilization of BSY as an alternative source of functional polysaccharides. BSY exhibited significantly higher carbohydrate content and lower crude protein content than SC and SB cells. The residues recovered through autolysis were 53.11%, 43.83%, and 44.99% for BSY, SC, and SB, respectively. Notably, the polysaccharide content of the BSY residue (641.90 μg/mg) was higher than that of SC (553.52 μg/mg) and SB (591.56 μg/mg). The yields of alkali-extracted water-soluble polysaccharides were 33.62%, 40.76%, and 42.97% for BSY, SC, and SB, respectively, with BSY comprising a comparable proportion of water-soluble saccharides made with SC and SB, including 49.31% mannan and 20.18% β-glucan. Furthermore, BSY demonstrated antioxidant activities, including superoxide dismutase (SOD), ABTS, and DPPH scavenging potential, suggesting its ability to mitigate oxidative stress. BSY also exhibited a significantly higher total phenolic compound content, indicating its potential to act as an effective functional food material.

## 1. Introduction

Yeast, a single-celled microorganism belonging to the fungal kingdom, has long been recognized for its pivotal role in various industrial processes, most notably in the production of alcoholic beverages such as beer [1,2,3]. Beside its application in fermentation processes, yeast has garnered scientific interest due to its rich composition, encompassing not only proteins extracted in the form of yeast extract, but also valuable polysaccharides residing within its cell wall [4,5].

Yeast cell walls are a source of useful polysaccharides with a range of biological activities and uses in medicine [1,3,6,7,8,9]. Past research has clarified the complex structure and composition of *Saccharomyces cerevisiae* cell walls, demonstrating that they make up 15–35% of the dry weight of the cell and are composed of up to 90% polysaccharides, organized in layers within the cell wall system [10]. These layers are typically arranged and consist of an inner rigid layer that is composed of chitin and alkali-insoluble glucans, the middle layer of alkali-soluble glucan, and an outer amorphous layer that is composed of phosphorylated mannoproteins [1]. Through (β1→3)- and (β1→6)-D-Glc connections (Glc—glucose), the β-glucans are connected. These polysaccharides impart a range of functional qualities to the cell, including immunomodulatory, antioxidative, and prebiotic benefits, while also maintaining its structural integrity [7,11].

The biomedical applications of yeast cell wall polysaccharides have been extensively explored, with promising outcomes achieved in various fields. β-glucans, renowned for their immunomodulatory properties, have shown potential in enhancing host defense mechanisms and combating infectious diseases [12]. Mannans, on the other hand, have garnered attention for their prebiotic effects and ability to promote gut health [13,14]. Additionally, chitin-derived compounds have exhibited wound healing properties and have been utilized in biomedical applications, such as tissue engineering and drug delivery systems.

Despite the biomedical potential of yeast cell wall polysaccharides, their utilization has primarily been restricted to niche applications, with yeast extract serving as the predominant source for industrial purposes. Brewer’s spent yeast (BSY), a by-product of beer production, represents a vast untapped reservoir of yeast cell wall polysaccharides. It is the second largest beer production by-product and significant quantities of BSY are disposed of annually, with current utilization limited to animal feed applications [2,15].

Given the rich polysaccharide content of yeast cell walls, BSY presents a promising, cost-effective alternative source of functional polysaccharides. This study investigates the composition and water-soluble cell wall polysaccharides extracted from BSY, comparing them with those taken from pure yeast cell cultures. By employing autolysis, a widely adopted and cost-effective technique, the yeast cells undergo disintegration. This yields yeast extract, a cytoplasmic component abundant in proteins, and autolysate residue primarily, comprising cell walls rich in polysaccharides [16,17]. Then, by elucidating the polysaccharide profiles of the cell wall, we aim to assess the feasibility of utilizing BSY-derived polysaccharides for various applications, including functional foods and nutraceuticals. Furthermore, we highlight the economic advantages of using BSY over preparing pure yeast cell cultures for polysaccharide extraction, emphasizing its viability, sustainability, and associated economic and environmental benefits. 

## 2. Materials and Methods

### 2.1. Materials

The Brewer’s spent yeast (BSY) utilized in this experiment was produced through drum drying by Korea Yeast Co., Ltd., (Mungyeong, Republic of Korea). The strain used was *Saccharomyces cerevisiae*. Pure strain cultures were prepared following the method described by Lee et al. (2023) [18] using *Saccharomyces cerevisiae* D452-2 (SC) and *Saccharomyces var. boulardii* ATCC MYA-796 (SB). Pre-cultures of the strains were grown at 30 °C and 250 rpm overnight in a YP medium containing 20 g of L^−1^ glucose. The pre-cultured cells were then harvested and inoculated into main cultures with an initial optical density at of 1.0600 nm. 

Fed-batch fermentation was conducted in a 2.5 L bioreactor (Kobiotech Co., Incheon, Republic of Korea) containing 1 L of YP medium with 20 g L^−1^ glucose and an initial OD 600 of 1.0. Upon the depletion of the initially added glucose, a feeding solution comprising 600 g L^−1^ glucose, 200 g L^−1^ yeast extract, and 100 g L^−1^ peptone was introduced at a rate of 15 mL h^−1^. The medium’s pH, temperature, agitation speed, and air supply were maintained at pH 5.5, 30 °C, 500 rpm, and 1.0 vvm, respectively. Following incubation, 60 to 70 g of cell dry matter was recovered and used for analysis. The yeast cells were washed twice with distilled water and subjected to centrifugation (8000 rpm, 10 min). Subsequently, the yeast cells were freeze-dried for further experimentation, and their weight was measured to calculate the yield.

### 2.2. Autolysis of Yeast Cell

Autolysis was performed using the method outlined by Tanguler et al. [16] in order to separate yeast into cell walls and extracts (Figure 1). Initially, 3% sodium chloride was added to yeast cell slurry (15% (*w*/*v*), pH 5.0) as a self-decomposition promoter, and the mixture was incubated for 24 h. Subsequently, the mixture was centrifuged at 5000 rpm, and the resulting residue was freeze-dried to obtain the cell wall fraction.

### 2.3. Water-Soluble Polysaccharide Extraction from Autolysate Residue

The isolation of water-soluble polysaccharides from autolysate residue (cell wall fraction) following autolysis was conducted based on the method described by [19] (Figure 1). Initially, 5 g of yeast cell wall was treated with 1% NaOH and extracted in a water bath at 100 °C for 2 h. The alkali-treated suspension was then centrifuged at 4500 rpm for 10 min, resulting in the separation of an insoluble polysaccharide pellet and a soluble polysaccharide supernatant. The supernatant solution pH was neutralized with 2 mol/L HCl and treated with cold ethanol at a ratio of 1:4. The mixture was subsequently precipitated at −80 °C for more than 2 h and then centrifuged at 8000 rpm for 10 min at −10 °C to recover the precipitate. This ethanol precipitation process was repeated two times to ensure thorough purification. Finally, the obtained precipitate was freeze-dried to yield a water-soluble polysaccharide fraction of yeast cells. 

### 2.4. Characterization of Yeast Cell Components

#### 2.4.1. Proximate Composition

The general compositions of BSY, SC, and SB cells and their respective autolysate residue were determined using the AOAC method [20]. Briefly, the moisture content was determined by oven drying method at 105 °C, the crude fat was assessed by the Soxhlet extraction method, the nitrogen was determined by the micro-Kjeldahl method and converted into crude protein via multiplication with the coefficient factor 6.25, and the crude ash was measured by heating the sample at 550 °C in a furnace. The value for carbohydrates was determined by adding the moisture content, crude fat, and crude protein to obtain a subtotal. The subtotal was subtracted from 100, and the remaining value represented the carbohydrate content. 

#### 2.4.2. Sugar Composition

To quantify sugar constituents, hydrolysis was performed with sulfuric acid based on the method in [21]. This acid hydrolysis released glucose from β-glucan, mannose from mannans, and glucosamine from chitin (since the *N*-acetyl residue in *N*-acetyl-glucosamine is acid-labile). We soaked 10 mg of samples in 75 µLof 72% (*w*/*v*) H_2_SO_4_ at room temperature for 3 h. Then, the slurry was diluted to 2 N H_2_SO_4_ by the addition of 0.95 mL MilliQ water containing 1 mg/mL Sorbitol (used as an internal standard to control the recovery) and heated at 100 °C for 4 h. Sulfate ions were neutralized by the drop-wise addition of NaOH until neutral pH values were reached. The volume was adjusted to 20 mL. We collected 1 mL of sample and centrifuged it at 13,000 rpm for 5 min. Then, the supernatant was filtered with a 0.22 μm syringe filter and used for analysis. High-performance anion exchange chromatography (HPAEC) was used to separate polysaccharides (as glucose, mannose, and N-acetyl-glucosamine). An ICS 5000 Dionex chromatography system (ICS-5000+, Thermo Fisher Scientific Co., Waltham, MA, USA) CarboPac PA100 column (250 × 4 mm, Dionex, Sunnyvale, CA, USA), a PEEK tube (0.24 mm i.d.), a gradient mixer (2 mm), an ED amperometry cell with a 0.25 µL channel volume, a pH-Ag/AgCl reference electrode, and gold electrodes were included. We used 100 mM sodium hydroxide and 1 M sodium acetate as analysis solvents, and samples were separated and analyzed under gradient conditions set to increase from 0 to 20 mM for 10–15 min and from 20 to 500 mM for 15–20 min. The column temperature was maintained at 40 °C, and the solvent flow rate was 1 mL/min.

#### 2.4.3. Gel Permeation Chromatography (GPC)

GPC was conducted to analyze the molecular weight distribution of water-soluble polysaccharides in each sample. First, the freeze-dried sample (10 mg/mL) was dissolved in distilled water and filtered using a 0.22 µm syringe filter (13 mm, 0.22 µL, Thermo Fisher Scientific Co.). The resulting filtered solution was utilized as the analysis sample. TSKgel G3000PW columns (7.8 mm × 30 cm, Tosoh, Tokyo, Japan) were employed in an HPLC system (Agilent Technologies, Inc., Santa Clara, CA, USA) for the analysis. Distilled water served as the mobile phase, with the column temperature maintained at 40 °C. The sample injection volume was 10 μL, and the solvent flow rate was set to 0.5 mL/min [22]. 

#### 2.4.4. Glycosidic Linkage Analysis Using ^1^H-NMR

To analyze the ⍺-1, 6 and ⍺-1, 2 glycosidic bonds between sugar molecules of water-soluble polymer samples, ^1^H-NMR spectroscopy (500 MHz FT-NMR, JEOL, Tokyo, Japan) was performed recovered via the alkali extraction of autolysate residue and ethanol precipitation. A freeze-dried sample of soluble polysaccharides (20 mg/mL) was dissolved in deuterium oxide (D_2_O) and reacted at 45 °C for 20 min. Then, ^1^H-NMR analysis was performed [22].

#### 2.4.5. FT-IR

FT-IR spectral analysis of soluble polysaccharides was carried out using the potassium bromide (KBr) pellet method with a Spectrum 3 FT-IR spectrophotometer (PerkinElmer Inc., Billerica, MA, USA) in the range of 400–4000 cm^−1^ [23].

### 2.5. Antioxidant Enzyme Activity of Water-Soluble Polysaccharide

#### 2.5.1. Superoxide Dismutase-like Activity (SOD)

We used the SOD assay kit and WST (Biomax, Seoul, Republic of Korea) for analysis, working according to the manufacturer’s instruction. The sample was dissolved in double-distilled water (DDW) at a concentration of 100 μL/mL. Next, 20 μL of the sample was added to the sample well and Blank 2 well on a 96-well plate. Additionally, 20 μL of DDW was added to the Blank 1 and Blank 3 wells. Following this, 200 μL of WST working solution was added to each well. To the Blank 2 and Blank 3 wells, 20 μL of dilution buffer was added, while to Blank 1 and the sample well, 20 μL of enzyme working solution was added. The reaction was then carried out at 37 °C for 20 min. 

Subsequently, the absorbance was measured at 450 nm using a microplate reader (Multiskan Sky, Thermo Scientific Co., Waltham, MA, USA).
$$\mathrm{SOD}\;\mathrm{activity}\;(\mathrm{inhibition}\;\mathrm{rate})\;(\%)=\frac{\left(B1-B3\right)-(S-B2)}{\left(B1-B3\right)}\times 100$$
where

B1: Blank 1/maximum absorbance.

B2: refers to blank 2/sample’s background absorbance.

B3: Blank 3/background absorbance of the rest of the solution except sample.

Sample: sample absorbance.

#### 2.5.2. ABTS Radical Scavenging

Following the method outlined by [24], ABTS and potassium persulfate were dissolved in distilled water to achieve final concentrations of 7.4 mM and 2.6 mM, respectively. This mixture was then stored in a dark place at room temperature for 18 h. Prior to use, it was diluted with phosphate-buffered saline (pH 7.4) to attain an absorbance of approximately 0.70 at 732 nm. Subsequently, 50 μL of the sample was combined with 1.0 mL of the diluted ABTS mixture and allowed to stand in darkness for 30 min. Following incubation, the absorbance was measured at 732 nm. Trolox, a standard antioxidant, was employed for calibration purposes, and the results were expressed as milligrams of Trolox equivalent per gram (mg Trolox equivalent/g) of the sample. Furthermore, the sample concentration (IC_50_), corresponding to a 50% scavenging rate, was calculated by establishing the relationship between concentration and scavenging rate.
$$\mathrm{ABTS}\;\mathrm{inhibition}\;(\%)=\frac{\left(C-S\right)}{C}\;\times 100$$where C refers to OD of control/black and S refers to OD of sample.

#### 2.5.3. DPPH Radical Inhibition

The DPPH radical’s scavenging ability, often employed as a representative indicator of antioxidant activity, was assessed following the method outlined by [25]. Initially, the sample was diluted to various concentrations, with each aliquot measuring 200 μL. Subsequently, 800 μL of a 0.4 mM DPPH solution was added to each sample, and the mixture was then incubated at 37 °C for 30 min. Following the incubation period, the absorbance was measured at 525 nm. The DPPH radical scavenging rate was determined by calculating the ratio of the decrease in absorbance observed in the sample treatment group in comparison to the absorbance of the control group (without sample).
$$\mathrm{DPPH}\;\mathrm{inhibition}\;(\%)=\frac{\left(C-S\right)}{C}\;\times 100$$
where C refers to OD of control/black and S refers to OD of sample.

#### 2.5.4. Total Phenolic Content

Using the method outlined by Horn [3] with minor modifications, the total phenol content was measured at 760 nm. This was achieved by adding 0.08 N Folin–Ciocalteu reagent to the sample and allowing it to stand at room temperature for 6 min. Subsequently, 3% Na_2_CO_3_ solution was added and left for 90 min. A standard curve was prepared using gallic acid as a standard reagent, and the total phenol content was expressed in mM gallic acid equivalent.

### 2.6. Statistical Analysis

The experimental results were presented as the average value and standard deviation (means ± SD) based on triplicate analyses. Significance testing between experimental groups was conducted using the SPSS statistical program (Ver. 23, Statistical Package for Social Sciences, SPSS Inc., Chicago, IL, USA), with the significance level set to *p* < 0.05.

## 3. Results and Discussion

### 3.1. Proximate Composition

The general protein, carbohydrate, and ash contents per dry weight of whole cell were ranged from 52.0 to 57.4%, 28.8 to 38.9%, and 6.4 to 8.5%, respectively (Table 1). Our findings align with previous studies [4,26,27,28], which reported similar parameters for yeast cells. Despite being rich in nutrients compared to plant-based food sources, BSY exhibited the lower protein, carbohydrate, and mineral contents than pure-cultured SC and SB. This disparity was attributed to SC and SB being cultured in YPD medium under constant conditions and recovered at the end of the stationary phase, at which point organelles are well-developed, resulting in higher nutrient content [29]. 

Unlike pure-cultured yeast, BSY is obtained after approximately 1 week of fermentation, during which it is composed of active and inactive yeast cells [3]. Additionally, BSY is mixed with wort and hop juice during fermentation. These factors likely contribute to the differences in composition observed between BSY and SC or SB. These comparative results provide valuable insights for the future utilization of cell debris residue as a food ingredient, particularly considering the lack of reports demonstrating the general composition of each strain of SC and SB when cultured at high concentrations.

The proximate composition of autolysate residue from BSY and pure-cultured cells (SC and SB) was determined to assess the effect of autolysis. Upon autolysis, the protein content of all samples decreased due to the removal of yeast cell internal components, which are protein-rich, such as yeast extract. Conversely, carbohydrate and mineral content increased, as the autolysate residue was primarily composed of cell wall. These results for BSY were consistent with the reported composition and shredded residue content of BSY. Contrary to the results observed for whole cells, the protein and ash contents of BSY autolysate residue were higher than those of cultured cells, while the carbohydrate content was lower. This difference in protein and ash content may be attributed to increases in cell wall mannoprotein and phosphate [30], as well as the adsorption of nutrients, originating from malt and hops, onto the cell wall during the brewing process [16].

### 3.2. Yield Components of Yeast Cell Fractionations

Yeast cell walls account for 15–30% of the yeast cell dry weight, of which up to 90% is composed of polysaccharides [1,31,32]. Table 2 shows the yield of 4 fractions of autolysis and extraction, including cell walls for BSY, SC, SB, and their polysaccharide content. Upon autolysis, BSY yielded lower autolysate, which refers to yeast extract/cytoplasmic components, and higher residue (constitute the cell wall) than SC and SB culture cells. This is mainly due to the nutrient-depleted beer manufacturing environment used for BSY as compared to that used to grow SC and SB in the fermenter. The findings of these autolysate fractions are consistent with a previous study [16] and suggest that the autolysate residue contains more than just cell wall components, as evidenced by a total polysaccharide content of approximately 600 µg/mg. 

To estimate the proportion of yeast cell wall, the composition of insoluble residue polysaccharides resulting from autolysis was determined. BSY exhibited the highest cell wall component content (34.09%), followed by SB (26.61%) and SC (24.25%), respectively. This disparity can be attributed to the higher concentration of β-glucan in the cell wall, likely resulting from modifications in cell wall polysaccharides prompted by the ethanol and osmotic stresses encountered during the brewing process [30].

The total carbohydrate, β-glucan, and mannan composition of the autolysate residue results align with previous studies on probiotic and brewer’s yeast cell wall composition under different growth conditions [29,33]. The values in this study indicated that β-glucans constituted 50–60% of the autolysate residue, while mannoproteins accounted for 35–40% [34]. Among the cell wall composition results, SB exhibited the highest mannan content (297.40 µg/mg), consistent with reports that SB, a lactic acid yeast, possesses a distinctive cell wall architecture with a thick mannan layer, resulting in higher mannan content [29].

These results suggest that BSY has comparable cell wall polysaccharide content to cultured SC and SB and serves as a rich source of functional cell wall polysaccharides, such as β-glucan and mannan. However, through alkali extraction, BSY yielded lower soluble polysaccharide sugar and higher insoluble residue compared to SC and SB, as described in detail in Section 3.3. According to Bastos et al. [30], the increase in insoluble polysaccharides, primarily comprising 1- 4 β-glucan, is linked to brewing-induced glycogen accumulation, thereby contributing significantly to the observed variance.

### 3.3. Characterization of Soluble Cell Wall Polysaccharides

#### 3.3.1. Soluble Cell Wall Polysaccharides Composition

Water-soluble polysaccharides (SP) were extracted from the autolysate residue using an alkaline solution, and their constituent saccharides, including β-glucan, mannan (mannoprotein), and chitin, were analyzed (see Table 3). SP contents were found to be higher in BSY (700.90 µg/mg), followed by SC (688.84 µg/mg) and SB (634.70 µg/mg), respectively. Upon detailed examination, a notably high mannan–glucan ratio was observed, consistent with previous studies on the probiotic makeup and cell wall composition of brewer’s yeast under glycerol cultivation conditions [29].

Research by Bastos et al. [30] demonstrated an increase in soluble β-glucan with the use of β-1,3 and β-1,6-D-Glc bonds derived from wort during brewing. This finding explains the higher proportion of water-soluble β-glucan in BSY (70:30) compared to SC (84:14) and SB (90:10). Additionally, alkaline-extracted fractions exhibited notably high levels of mannan, known for its water solubility compared to other yeast cell wall components, such as β-glucan and chitin [6,35,36]. Given mannan’s immune-enhancing properties, this mannan-rich soluble polysaccharide (mannan: β-glucan 70:30) holds significant potential in the nutraceutical industries. Furthermore, the alkali-insoluble fraction predominantly comprises β-glucan, as reported by [29] (see Appendix A).

#### 3.3.2. Molecular Weight Determination

The molecular weight size distribution of soluble polysaccharides (SP) obtained from each yeast autolysate residue was assessed via gel permeation chromatography (GPC), as illustrated in Figure 2. Previous research indicates that SPs extracted from yeast cell walls typically exhibit molecular weights ranging from 166 to 700 kDa [22]. Notably, the total weight of the average molecular weight (Mw) of SPs derived from spent yeast (185.8 kDa) was significantly lower than that of the cultured SC and SB strains, measuring 302.9 kDa and 261.7 kDa, respectively (Table 4). Further analysis revealed the molecular weight of the height peak (Mp) for SC-SP and SB-SP to be 452.5 kDa and 405.3 kDa, respectively, while BSY-SP exhibited a notably lower Mp of 19.3 kDa. The subsequent confirmation of the degree of polymerization (DP) at Mp unveiled values of 2793.2 for SC-SP, 2502.1 for SB-SP, and 118.9 for BSY-SP, indicating a higher degree of polymerization in SC-SP.

The presence of a large quantity of water-soluble glucose in BSY-SP likely influences its molecular weight, contributing to its lower size compared to other polysaccharides. Numerous studies have highlighted that polysaccharides with smaller molecular weights tend to possess simpler structures and display enhanced water solubility, thus exhibiting heightened biological activity [37,38]. 

A comparison between strains (SC and SB) revealed distinct characteristics in their polysaccharide fractions. SC exhibited a greater proportion of polymeric fractions with a size of around 400 kDa, whereas SB displayed a slightly higher content of polysaccharides with sizes of approximately 22 kDa and above. This aligns with previous findings, indicating that SB strains typically feature a thicker mannan layer, a characteristic component of the outer membrane.

#### 3.3.3. ^1^H-NMR Spectroscopic Identification

The results illustrating the confirmation of the intermolecular binding form of water-soluble polysaccharide (SPs) extracted from the cell wall are depicted in Figure 3. Anomeric H atoms, crucial for identification, resonate within the range of 4.9–5.5 ppm, while the remaining H atoms fall within 3.5–4.5 ppm, as per a previous study Kath and Kulicke [22]. However, due to numerous overlapping signals in this range, precise assignment is unfeasible. The assignment of anomeric H atoms relied upon published data for higher mannan oligosaccharides, which were further corroborated by methylation analyses and other findings [39]. Notably, the signal observed at 5.1–5.2 ppm (positions A) signifies the presence of α-1,6-mannans in the basic chain, to which the mannose side groups are attached. Additionally, a signal corresponding to an anomeric H atom of a terminal α-1,3-linked mannose is discernible in this region (position D). Another distinctive signal at 5.3 ppm is indicative of the presence of mannose in α-1,2-linked side chains composed of two or more sub-units (position C). Furthermore, the signal at 5.04 ppm denotes terminal α-1,2-linked mannose and α-1,2-linked mannose with a mannose substituent in the 3-position (position B, D). Occasionally, a band appears at 5.41 ppm (positions E), attributed to mannose bonding to the mannan complex via a phosphodiester bridge [22].

BSY-SP exhibited a characteristic peak at a chemical shift of 5.41 ppm, likely arising from the formation of complexes due to interactions with residual substances present in the wort during beer fermentation. Furthermore, the intensity of this peak tended to be relatively lower compared to that of SC-SP and SB-SP. This difference is believed to stem from the mixture of various components during the brewing process and the lower purity inherent in yeast cells cultured in a non-controlled environment.

Tamano et al. [40] reported that the soluble form of beta-glucan typically manifests a peak at a chemical shift of 4.5 ppm, attributable to beta-1,6-linked glucose bonds. Additionally, Chen et al. [41] noted in a structural determination study involving the hydrolysis of barley that the peak corresponding to 5.3 ppm in the spectral results of 1–3- or 1–4-linked beta-glucan, assessed via ^1^H-NMR, could be attributed to residual anomeric protons of the 1,4-glucoside bond within beta-glucan.

#### 3.3.4. FT-IR Spectroscopic Identification

The FT-IR spectrum results of cell wall-soluble polysaccharides (Figure 4) exhibit typical cell wall polysaccharide patterns, consistent with findings from various studies [6,31,42,43,44]. A broad peak spanning 3650–3200 cm^−1^ corresponds to O-H stretching bonds, indicative of changes in bond length with hydroxyl groups, while a weaker bend observed between 3000 and 2800 cm^−1^ is associated with C-H stretching bonds in carbonyl groups [45,46]. Takallo et al. [17] suggested that the wavelength range of 3650–3200 cm^−1^, representing hydroxyl groups, may influence the formation or alteration of intramolecular and intermolecular hydrogen bonds, or cause variations in the polymerization of cell wall polysaccharides or yeast cell wall polysaccharides, affecting their helical structure.

Moreover, a peak observed between 2985 and 3015 cm^−1^ is attributed to the presence of lipid residues within the cell wall [47]. The regions spanning 1700–1500 cm^−1^ are associated with C=O asymmetric and symmetric stretching bonds, while a peak at 1374 cm^−1^ corresponds to C-H bending bonds, where changes in bond angles occur [48]. Additionally, a peak between 1155 and 1080 cm^−1^ signifies a C-O bending bond, while the presence of a C-O stretching bond is indicated by a peak at 1024 cm^−1^ [27]. Oscillations between 520 and 1100 cm^−1^ are linked to various polysaccharides (α-glucan, β-glucan, α-mannan, etc.) and can be subdivided into the sugar region (950–1200 cm^−1^) and the anomeric region (750–950 cm^−1^) [49].

Notably, in SB-SP, the hydroxyl groups at the 3650–3200 cm^−1^ peak appear prominent, suggesting the predominance of O-H stretching bonds, which is likely due to the thick mannoprotein layer characteristic of SB strains [50]. C-O bond stretching, indicated by the 1024 cm^−1^ peak, is more pronounced in SC-SP and SB-SP compared to BSY-SP. Within the range of 700–1500 cm^−1^, large peaks corresponding to C=O asymmetric and symmetric stretching bonds, as well as C-H bending bonds at 1374 cm^−1^, are observed in BSY-SP, indicative of β-glucan presence. Additionally, the intense absorption of the C=O stretching peak was attributed to the presence of double-bonded substances such as hop acids, derived from the wort and hop juice mixtures used in the brewing process [51]. Moreover, the strong absorption peak at 810 cm^−1^ suggests the presence of α-anomeric configurations in the polysaccharides [45]. The comparison of the 810 cm^−1^ peak areas reveals that SC-SP and SB-SP exhibit more α-anomeric configurations and a higher percentage of mannose compared to BSY-SP.

#### 3.3.5. Antioxidant Enzyme Activity of Water-Soluble Polysaccharide

The water-soluble polysaccharide components of yeast cells exhibited antioxidant activity against superoxide, ABTS, and DPPH radicals (Table 5). The results indicate that brewers’ spent yeast polysaccharides displayed significantly higher antioxidant activity compared to the soluble polysaccharides taken from cultured yeast cells. This disparity may be attributed to the presence of phenolic compounds, such as hop acids, derived from the beer manufacturing process. Previous studies have shown that hop acids from beer bind to yeast cells and only become soluble at higher pH levels [16]. There is substantial evidence suggesting that hop acids possess antioxidant properties [51]. Further analysis of total phenolic acid (TPA) content supports this finding, revealing that the soluble polysaccharide fraction of brewers’ spent yeast contains three times more TPA content. Additionally, Figure 5 also indicates that BSY polysaccharides exhibited significantly lower IC_50_ values than pure-cultured polysaccharides.

Previous research on the antioxidant activity of yeast polysaccharides has indicated that the insoluble β-glucan exhibits lower antioxidative activity compared to the alkali-soluble mannan and β-glucan [52]. The authors of this study concluded that the protein residue present on the mannan contributes significantly to its antioxidative properties. However, it is noteworthy that β-glucan derived from malt also possesses considerable radical scavenging potential [53]. This finding might contribute to explaining the higher antioxidant activities observed in brewers’ spent yeast compared to the pure-cultured yeast strains SC and SB, which also contain higher β-glucan contents.

## 4. Conclusions

Based on the findings, it is evident that BSY represents a promising alternative to cultured yeast cells due to its enriched nutrient profile and functional polysaccharide composition. Its autolysis residue, predominantly composed of the cell wall, demonstrates higher polysaccharide content, particularly β-glucan and mannan, than that of pure-cultured SC and SB. Furthermore, although it is mannan-rich in all yeasts, the soluble polysaccharide composition of BSY reveals elevated levels of β-glucan and reduced mannan concentration compared to SC and SB. Moreover, these soluble extracts exhibit superior antioxidant activity, as evidenced by their higher scavenging potential against SOD, ABTS, and DPPH radicals compared to cultured SC and SB yeast cells, attributable in part to the presence of hop acid residue, which offers an additional advantage. Overall, BSY emerges as a cost-effective and sustainable source of functional soluble polysaccharides, offering a viable alternative to cultured yeast cells as a food ingredient. This also contributes to waste reduction in beer production processes. These findings underscore the potential of BSY to enhancing the nutritional and functional aspects of food products, warranting further exploration and utilization in various food formulations and applications.

## Figures and Tables

**Figure 1 foods-13-01567-f001:**
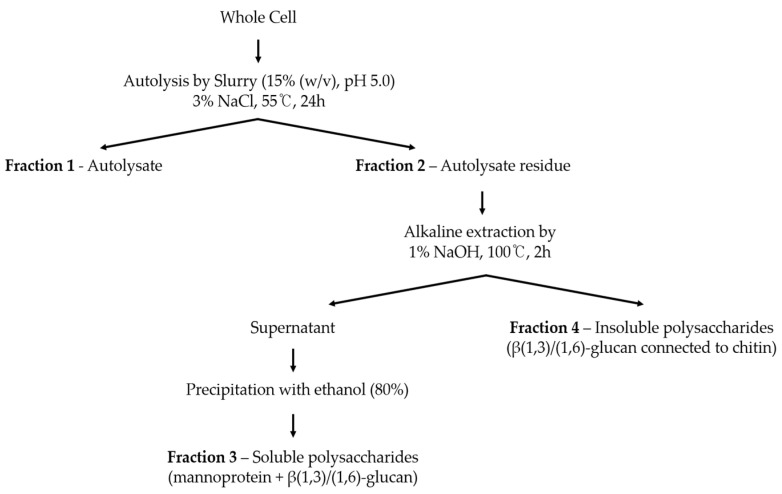
Schematic diagram of the fractionation of soluble polysaccharides from yeast cell. F1—autolysate. This refers to the cytoplasmic components of the cell or yeast extract. F2—autolysate residue. This refers to the insoluble component of yeast cells composed of mainly cell wall. F3—soluble polysaccharides. This refers to alkali-soluble components extracted from autolysate residue. F4—insoluble polysaccharides. This refers to the alkali-insoluble components of autolysate residue.

**Figure 2 foods-13-01567-f002:**
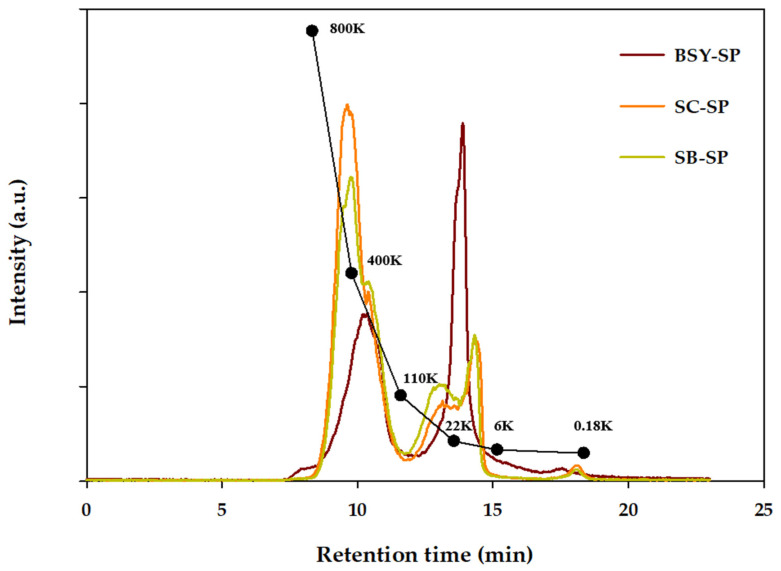
Gel permeation chromatography analysis of yeast cell water-soluble polysaccharide. BSY-SP, Brewer’s spent yeast-soluble polysaccharide; SC-SP, *S. cerevisiae*-soluble polysaccharide; SB-SP, *S. boulardii*-soluble polysaccharide.

**Figure 3 foods-13-01567-f003:**
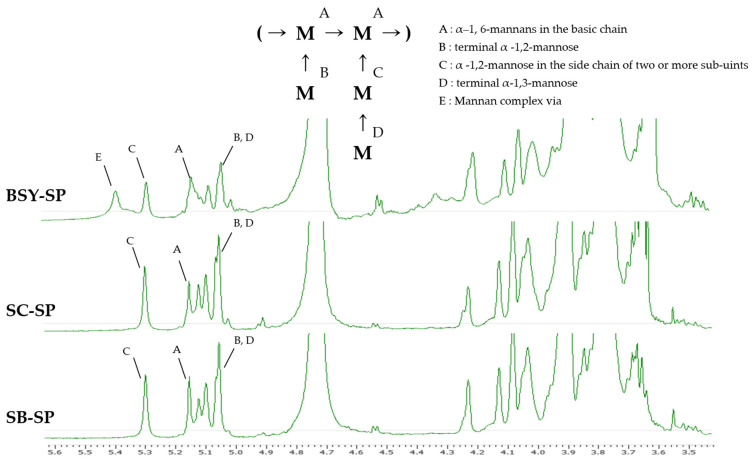
^1^H-NMR spectrum of the soluble polysaccharide.

**Figure 4 foods-13-01567-f004:**
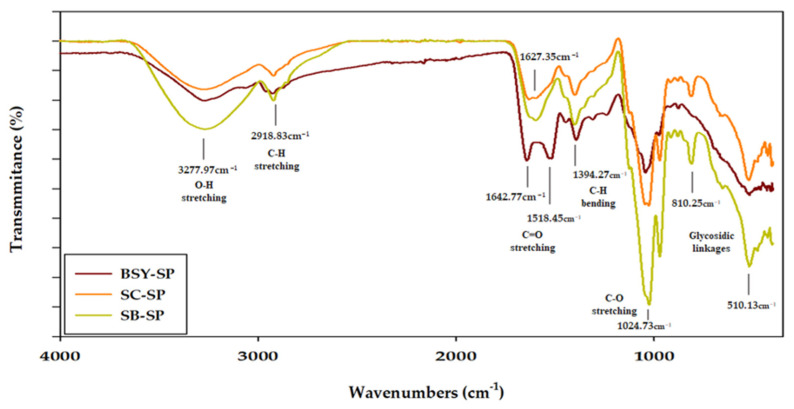
FT-IR spectrum of yeast cell-soluble polysaccharide. BSY-SP, Brewer’s spent yeast-soluble polysaccharide; SC-SP, *S. cerevisiae*-soluble polysaccharide; SB-SP, *S. boulardii*-soluble polysaccharide.

**Figure 5 foods-13-01567-f005:**
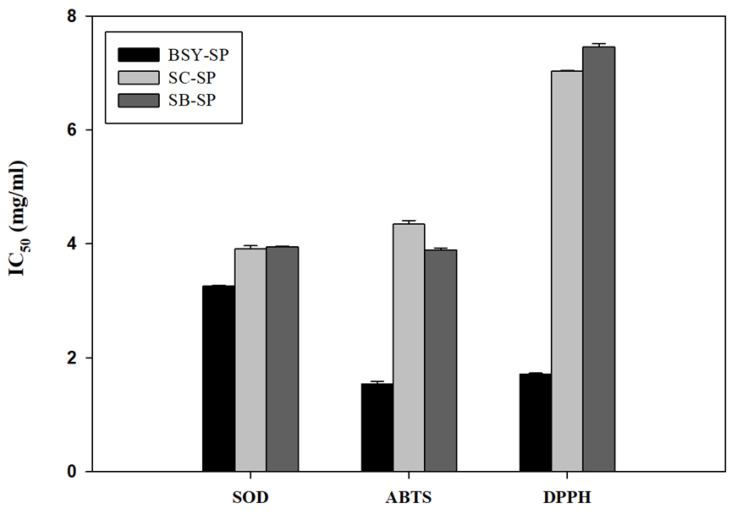
Antioxidant activities of soluble polysaccharides derived from autolyzed yeast cell residue.

**Table 1 foods-13-01567-t001:** Proximate composition of whole and autolyzed cell residue from BSY, *S. cerevisiae* and *S. boulardii*.

Components (%)	BSY		*S. cerevisiae*		*S. boulardii*	
	WC	AR	WC	AR	WC	AR
Crude protein	52.04 ± 1.98 ^c^	30.50 ± 0.21 ^a^	53.46 ± 2.43 ^b^	18.64 ± 0.06 ^c^	57.43 ± 2.71 ^a^	19.57 ± 0.38 ^b^
Carbohydrate	28.80 ± 0.54 ^c^	53.13 ± 0.23 ^c^	38.87 ± 1.25 ^a^	67.17 ± 0.09 ^a^	33.65 ± 0.95 ^b^	64.65 ± 0.41 ^b^
Ash	6.43 ± 0.04 ^c^	10.39 ± 0.06 ^a^	7.35 ± 0.04 ^b^	7.77 ± 0.01 ^c^	8.47 ± 0.07 ^a^	8.39 ± 0.04 ^b^

AR: autolysate residue, WC: whole cell. Values are means ± SD (n = 3). Mean comparisons were made between WC and AR and different superscript letters in the same row indicates the significant difference at *p* < 0.05.

**Table 2 foods-13-01567-t002:** Yields of yeast cell autolysis fractions and their polysaccharide composition.

Components	BSY	*S. cerevisiae*	*S. boulardii*
Autolysate (%) (F1)	39.67 ± 0.03 ^c^	55.06 ± 0.08 ^a^	50.82 ± 0.04 ^b^
Autolysate residue (%) (F2)	53.11 ± 0.07 ^a^	43.83 ± 0.07 ^c^	44.99 ± 0.05 ^b^
*β-glucan* (μg/mg)	346.21 ± 15.12 ^a^	284.58 ± 10.03 ^b^	268.53 ± 11.85 ^c^
*Mannan* (μg/mg)	291.73 ± 5.75 ^a^	255.26 ± 4.76 ^b^	297.40 ± 10.91 ^a^
*Chitin* (μg/mg)	3.96 ± 0.04 ^c^	13.68 ± 0.54 ^b^	25.63 ± 1.25 ^a^
*Total* (μg/mg)	641.90 ± 6.28 ^a^	553.52 ± 4.66 ^c^	591.56 ± 2.32 ^b^
*Cell wall % (on cell dry mass)*	34.09 ± 0.12 ^a^	24.25 ± 0.03 ^c^	26.61 ± 0.09 ^b^
Soluble polysaccharides (F3)	33.62 ± 0.09 ^c^	40.76 ± 0.14 ^b^	42.97 ± 0.21 ^a^
Insoluble polysaccharides (F4)	44.14 ± 0.18 ^a^	31.17 ± 0.26 ^c^	33.76 ± 0.23 ^b^

Values in italics were polysaccharides determined from autolysis insoluble residue (F2). Data in the same row with different letters represent significant differences at *p* < 0.05.

**Table 3 foods-13-01567-t003:** Saccharides composition of soluble polysaccharide derived from autolyzed yeast cell residue in (µg/mg).

Polysaccharides	BSY	*S. cerevisiae*	S. *boulardii*
β-glucan	201.80 ± 5.41 ^a^	87.89 ± 2.42 ^b^	56.21 ± 2.70 ^c^
Mannan	493.11 ± 7.45 ^c^	596.25 ± 10.25 ^a^	573.75 ± 7.45 ^b^
Chitin	5.99 ± 0.13 ^a^	4.70 ± 0.01 ^c^	4.74 ± 0.01 ^b^
Total	700.90 ± 1.50 ^a^	688.84 ± 4.07 ^b^	634.70 ± 4.43 ^c^
Mannan: β-glucan	70:30	86:14	90:10

Data in the same row with different letters represent significant difference (*p* < 0.05).

**Table 4 foods-13-01567-t004:** Molecular weight distributions of water-soluble polysaccharide.

Sample	Mw ^1^	Mp ^2^	DP ^3^
BSY	185.8	19.3	118.9
*S. cerevisiae*	302.9	452.5	2793.2
*S. boulardii*	261.7	405.3	2502.1

^1^ Mw; total weight average molecular weight (kDa); ^2^ Mp; molecular weight of highest peak (kDa); ^3^ DP; degree of polymerization. DP was determined by dividing the Mp value by the molecular weight of one glucose molecule (162 Da).

**Table 5 foods-13-01567-t005:** Antioxidant enzyme activity of soluble polysaccharide.

Samples	SOD Activity (%)	ABTS Scavenging µgTE/mg	DPPH Inhibition µgTE/mg	TPA mgGAE/g
BSY-SP	76.35 ± 1.75 ^a^	192.38 ± 2.77 ^a^	535.01 ± 4.20 ^a^	81.19 ± 1.56 ^a^
SC-SP	63.81 ± 1.25 ^b^	86.70± 4.93 ^b^	359.79 ± 2.26 ^b^	43.08 ± 1.69 ^b^
SB-SP	62.55 ± 2.32 ^b^	78.99± 1.52 ^c^	306.54 ± 2.74 ^c^	49.63 ± 0.64 ^c^

SC-SP, *S. cerevisiae*-soluble polysaccharides; SB-SP, *S. boulardii*-soluble polysaccharides; BSY-SP, Brewer’s spent yeast-soluble polysaccharides., TPA: total phenolic acid presented as mg of gallic acid equivalent per gram of sample. ABTS and DPPH activity were presented as micrograms of Trolox equivalent per mg of polysaccharide samples. Values are means ± SD (n = 3). Means with different letters in the same column are significantly different at *p* < 0.05.

## Data Availability

The original contributions presented in the study are included in the article/Appendix A, further inquiries can be directed to the corresponding author.

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
