# Peer review of "A Comparative Study of Composition and Soluble Polysaccharide Content between Brewer’s Spent Yeast and Cultured Yeast Cells"

_foods, 2024, doi:10.3390/foods13101567_

Round 1
Reviewer 1 Report
Comments and Suggestions for Authors
The manuscript ‘A Comparative Study of Composition and Soluble Polysaccharide content between Brewer’s Spent Yeast and Cultured Yeast Cell’ is complete, the topic is interesting and should be of interest to readers of Foods. I only have two small comments:
79-80 – The name Saccharomyces boulardii should be in italics
Please provide literature references to the methodologies given in sections 2.2, 2.3, 2.4 and 2.5.
Author Response
Dear Reviewer
Please find the attached file on the response to our manuscript.

Reviewer 2 Report
Comments and Suggestions for Authors
The paper is well-written, well-organized and well-referenced. The study's premise is clearly established, and the whole experimental setup is clearly described. Only minor revisions are necessary. Therefore, in this reviewer’s opinion, the manuscript needs to take into account some of the comments and responds to some questions.
All my suggestions and questions for the authors are provided in the attached file. I hope the comments are constructive and help the authors to improve the manuscript.

Author Response
Dear reviewer
Please find attached the response to your comments and questions on our manuscript.
Warm regards!
